# Uncovering Genomic Features and Biosynthetic Gene Clusters in Endophytic Bacteria from Roots of the Medicinal Plant *Alkanna tinctoria* Tausch as a Strategy To Identify Novel Biocontrol Bacteria

Henry D. Naranjo,[a]* Angélique Rat,[a]§ Noémie De Zutter,[b] Emmelie De Ridder,[a] Liesbeth Lebbe,[a] Kris Audenaert,[b] Anne Willems[a]

[a]Laboratory of Microbiology, Department of Biochemistry and Microbiology, Faculty of Sciences, Ghent University, Ghent, Belgium
[b]Laboratory of Applied Mycology and Phenomics, Department of Plants and Crops, Faculty of Bioscience Engineering, Ghent University, Ghent, Belgium

Henry D. Naranjo and Angélique Rat contributed equally to this work and share the first authorship. Henry D. Naranjo is listed first, as he was responsible for coordinating the analysis and writing.

**ABSTRACT** The world's population is increasing at a rate not seen in the past. Agriculture, providing food for this increasing population, is reaching its boundaries of space and natural resources. In addition, changing legislation and increased ecological awareness are forcing agriculture to reduce its environmental impact. This entails the replacement of agrochemicals with nature-based solutions. In this regard, the search for effective biocontrol agents that protect crops from pathogens is in the spotlight. In this study, we have investigated the biocontrol activity of endophytic bacteria isolated from the medicinal plant *Alkanna tinctoria* Tausch. To do so, an extensive collection of bacterial strains was initially genome sequenced and *in silico* screened for features related to plant stimulation and biocontrol. Based on this information, a selection of bacteria was tested *in vitro* for antifungal activity using direct antagonism in a plate assay and *in planta* with a detached-leaf assay. Bacterial strains were tested individually and in combinations to assess the best-performing treatments. The results revealed that many bacteria could produce metabolites that efficiently inhibit the proliferation of several fungi, especially *Fusarium graminearum*. Among these, *Pseudomonas* sp. strain R-71838 showed a strong antifungal effect, in both dual-culture and *in planta* assays, making it the most promising candidate for biocontrol application. Using microbes from medicinal plants, this study highlights the opportunities of using genomic information to speed up the screening of a taxonomically diverse set of bacteria with biocontrol properties.

**IMPORTANCE** Phytopathogenic fungi are a major threat to global food production. The most common management practice to prevent plant infections involves the intensive use of fungicides. However, with the growing awareness of the ecological and human impacts of chemicals, there is a need for alternative strategies, such as the use of bacterial biocontrol agents. Limitations in the design of bacterial biocontrol included the need for labor-intensive and time-consuming experiments to test a wide diversity of strains and the lack of reproducibility of their activity against pathogens. Here, we show that genomic information is an effective tool to select bacteria of interest quickly. Also, we highlight that the strain *Pseudomonas* sp. R-71838 produced a reproducible antifungal effect both *in vitro* and *in planta*. These findings build a foundation for designing a biocontrol strategy based on *Pseudomonas* sp. R-71838.

**KEYWORDS** bacterial genomics, biocontrol, secondary metabolites

The growth of the world's population is pushing agriculture to its limits. Natural resources are being consumed at a speed that does not allow their regeneration, contributing to climate change, declining biodiversity, and the general loss of ecosystem services (1). As

**Editor** Frédérique Reverchon

Address correspondence to Anne Willems, Anne.Willems@UGent.be.

*Present address: Henry D. Naranjo, Facultad de Ciencias Químicas, Universidad Central del Ecuador, Quito, Ecuador.

§Present address: Angélique Rat, Bergelson Lab, Department of Biology, New York University, New York, New York, USA.

The authors declare no conflict of interest.

agriculture is one of the main polluters, practices are being adapted to reduce impacts on the environment and human health. Part of this strategy is to replace agrochemicals with nature-based solutions; in this context, microbial inoculants are gaining importance as biostimulants or biocontrol agents.

A major agricultural threat is the diseases caused by plant-pathogenic fungi, viruses, and bacteria, with about 16% of the world's crop losses due to microbial diseases and 70 to 80% of these losses caused by fungi (2). The most common management practice to prevent or combat diseases involves using pesticides. However, the excessive application of agrochemicals has undesirable effects on nontarget organisms and causes environmental and human health concerns, making the use less accepted by the public (3, 4). An alternative strategy is using microbes that can control these diseases, the so-called biocontrol organisms. The application of biocontrol organisms aims to protect plant health with a minimal impact on human health and the environment. The market is expected to increase by 15% annually in the next 5 years (5). Current applied biocontrol strategies often rely on well-known genera, such as *Pseudomonas* and *Bacillus*, and comprise one single strain against a specific pathogen. However, only a few studies have been performed so far on the testing of taxonomically diverse bacteria and on the use of microbial consortia or synthetic communities (SynComs) to control phytopathogenic fungi. Studying underexplored genera and using bacterial SynComs as biocontrol agents represents a promising approach toward the discovery of novel and effective control of phytopathogens (6–9).

The modes of action for biocontrol bacteria are broad. They comprise the induction of systemic resistance (ISR) in plants, competition for an ecological niche or a substrate, production of lytic enzymes, and production of inhibitory allelochemicals (10). The induction of local and systemic plant defenses by beneficial bacteria typically involves the activation of jasmonic acid, salicylic acid, ethylene, or abscisic acid signaling pathways in plants (11). Especially rhizobacteria belonging to the genera *Pseudomonas* and *Bacillus* are well known for their ability to trigger ISR. A second mode of action is competition for space and nutrients. A typical example of competition is the production of siderophores. These are low-molecular-weight compounds with a high affinity for ferric iron (12, 13). Siderophores act as antimicrobials by causing iron starvation to non-siderophore-producing or poorly siderophore-producing microorganisms. A third way microorganisms can compete with plant pathogens is by producing enzymes hydrolyzing a wide variety of polymeric compounds. Finally, antibiosis through the production of antimicrobial compounds is an effective strategy for controlling phytopathogenic fungi. An example of this mechanism of action is represented by *Pseudomonas chlororaphis* strain PA23, which showed an antimicrobial effect against the plant pathogen *Sclerotinia sclerotiorum* by producing volatile and nonvolatile antibiotics (14).

The exploration of underscrutinized bacterial habitats, such as sediments, insects, or plants, is encouraged in order to discover antimicrobial compounds and competent microorganisms to be used as biocontrol agents (15). The close relationship between plants and bacteria, which often involves interkingdom communication, is likely to entail the production of molecules with diverse bioactive potential (16). The biological association of plants and bacteria as the source of bioactive molecules is even more interesting in the case of plants already known for medicinal purposes. Despite several studies highlighting the antimicrobial potential of bacteria isolated from medicinal plants, this area of research is still largely underexplored (17–21).

In this work, we aimed to study the effect of a highly diverse range of bacteria, previously isolated from the medicinal plant *Alkanna tinctoria*, for their biocontrol effects against several phytopathogenic fungi. To achieve this, bacterial genomes were sequenced and screened *in silico* to provide a bioprospection framework for the efficient selection of bacterial biocontrol candidates. These were subsequently tested using a dual-culture assay against important phytopathogens that affect agriculture worldwide, including the fungi *Rhizoctonia solani*, *Fusarium oxysporum*, *Fusarium graminearum* and the phytopathogenic oomycete *Pythium ultimum*. Then, an assay supported by a multispectral imaging system was used to evaluate the effect of single bacteria and SynComs against *Fusarium graminearum* PH-1 in wheat detached leaves.

## RESULTS

**Mining of genomic features for plant growth promotion (PGP) and biocontrol activity.** In this work, we aimed to identify biocontrol properties in a diverse selection of bacteria isolated from a medicinal plant based on their genomic features. First, a whole-genome sequence (WGS) analysis was performed on 88 strains, followed by annotation of biosynthetic gene clusters (BGCs). As a first step toward identifying strains with abundant BGCs, we used Illumina technology for WGS. All genomes were found to be unique (pairwise average nucleotide identity [ANI] values below 99%). For a few selected strains with abundant BGCs (>10 BGCs, with truncated or redundant elements based on our manual inspection of the antiSMASH results or with highly fragmented genomes), we complemented the sequencing step with the sequencing of long reads using nanopore technology. This strategy allowed us to obtain closed and full genomes that produced intact BGCs predicted by antiSMASH. A full list of the genomes and details are shown in Table S1 in the supplemental material.

Our approach uncovered the potential of specific bacterial groups to produce compounds with antimicrobial or antifungal activity. BGCs encoding nonribosomal peptides (NRPs), polyketides (PKS), and ribosomally synthesized and posttranslationally modified peptides (RiPPs) were abundant and highly relevant in this context. Interestingly, our results revealed that strains from the classes *Gammaproteobacteria*, *Chitinophagia*, and *Bacilli* were enriched in these types of BGCs (Fig. 1). Among these, *Chitinophaga* sp. R-72609 presented the highest number of annotated BGCs of the whole collection, with 24 biosynthetic clusters, followed by several *Pseudomonas* sp. (16 or more clusters), *Chitinophaga* sp. R-73072 (16 clusters), *Mycobacterium* sp. R-73050 (16 clusters), *Micromonospora* sp. R-74116 (14 clusters) and *Brevibacillus* sp. R-71971 (14 clusters). While *Pseudomonas* and *Brevibacillus* species are known and used in biocontrol applications, *Chitinophagia*, *Mycobacterium*, and *Micromonospora* are largely underexplored and represent thus a great interest for future studies.

To provide additional relevant information for future studies on plant-microbe interactions, genes and pathways related to plant growth-promoting properties such as the biosynthesis of 1-aminocyclopropane-1-carboxylic acid (ACC) deaminase and indole-3-acetic acid (IAA), phosphate solubilization, and endophytic colonization (i.e., cellulose and pectin metabolism and secretion systems) were also annotated in the genomes (Fig. 1; File S1). Most of these results are in accordance with the PGP *in vitro* assay results from our previous study (22).

**Dual-culture assay as an *in vitro* approximation to detect biocontrol activity.** Following the genome analysis and the identification of potential biocontrol features that involve the production of metabolites, a subset of 63 bacteria was selected and tested for biocontrol activity using a dual-culture assay against *Rhizoctonia solani* MUCL 9418, *Pythium ultimum* MUCL 53834, *Fusarium oxysporum* MUCL 781, and *Fusarium graminearum* PH-1. This selection included 7 strains rich in BGCs (16 or more BGCs), 13 strains randomly chosen out of 19 with an intermediate number of BGCs (between 10 and 15 BGCs), and 43 strains randomly selected out of 61 that were poor in BGCs (fewer than 10 BGCs). The full list of strains and information are presented in Table 1.

Twelve bacterial strains, *Pseudomonas* sp. R-71838, R-71842, R-72102, R-72172, R-72008, and R-72210, *Inquilinus* sp. R-72501, *Luteibacter* sp. R-72151 and R-73110, *Bacillus cereus* R-74298, *Variovorax* sp. R-72016, and *Brevibacillus* sp. R-71971, strongly reduced the growth of three fungi in the dual-culture assay (Table 1, categories A, B, and C). Six strains inhibited at least one of the fungi, while 45 strains showed a weak or no effect against the fungi (Table 1, categories D and E). *Pseudomonas* and *Brevibacillus* strains inhibited the growth of one or several fungi at a distance (Fig. 2, categories A and B), likely producing secondary metabolites that were able to diffuse through the culture medium. In contrast, the mechanism of the *Inquilinus* sp., *Variovorax* sp., and *Luteibacter* sp. involved spreading or surrounding the fungi and inhibiting mycelial expansion at close contact (Fig. 2, category C). *Inquilinus* is a genus previously reported from human clinical samples and from soil, and our endophytic strain showed the potential to control phytopathogenic fungi for the first time. Moreover, in the presence of *Pseudomonas orientalis* R-71842, the mycelium color of *F. oxysporum* MUCL 781 changed from pink to green, *R. solani* MUCL 9418 changed from brown to pink, and

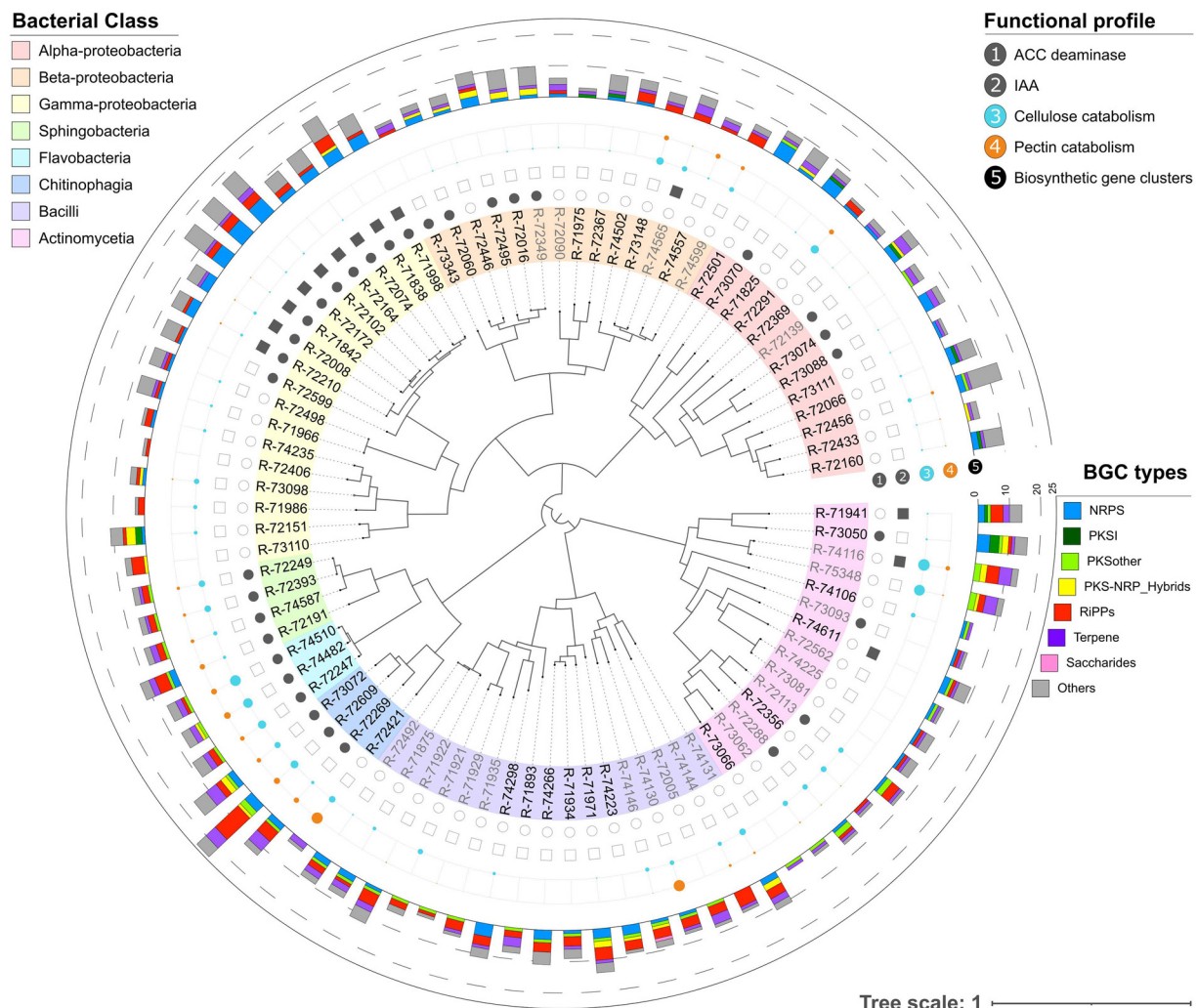

**FIG 1** Phylogenomic tree representing bacterial strains used for whole-genome sequencing. Branch tips contain strain codes in black (bacteria selected for antifungal *in vitro* screening), nonselected strains are depicted in gray, and background colors represent the different taxonomic groups (bacterial class) they belong to. Traits relevant for plant-microbe interactions are shown in layers 1 to 5. Layers 1 and 2 indicate the presence (circle or square, solid color) or absence (circle or square, noncolored) of annotated enzymes/pathways for 1-aminocyclopropane-1-carboxylic acid (ACC) deaminase and indole-3-acetic acid (IAA), respectively. Layers 3 and 4 represent the abundance of genes for cellulose and pectin-degrading enzymes; the circle size reflects the number of genes. Finally, layer 5 represents the number of biosynthetic gene clusters (BGCs) distributed by types according to the different colors.

*P. ultimum* MUCL 53834 changed from white to brown, which points to an interference with fungal secondary metabolism. Interestingly, some bacteria, such as *Pantoea agglomerans* R-71966, showed specific antifungal activity depending on the fungus tested, with a good effect against *F. oxysporum* MUCL 781, no effect against *R. solani* MUCL 9418, and a weak effect against *P. ultimum* MUCL 53834.

Results from the dual-culture assay confirmed that several strains enriched in BGCs exert a biocontrol effect against phytopathogenic fungi (Table 1 and Fig. 2, categories A and B). Four of the seven strains, in the subset defined as rich in BGCs (*Pseudomonas* sp. R-71838, R-71842, R-72102, and R-72172), produced metabolites reducing fungal growth. Furthermore, 5 out of 13 bacteria selected with an intermediate amount of BGCs showed an antifungal effect. Finally, out of the 43 bacteria with low BGC counts, only 9 showed an antifungal effect.

**Detached-leaf assay as a tool to test the efficiency of the biocontrol effect in wheat leaves.** Several of the strains selected based on their genomic features showed strong antifungal activity in the dual-culture assay (Table 1). These strains were also tested using the detached-leaf assay against *F. graminearum* PH-1, as well as strains that were not highlighted by the genomic study but that showed an interesting effect in the dual-culture

**TABLE 1** Effects of 63 bacterial strains on the fungal growth (dual-culture assays) of *Fusarium oxysporum* MUCL 781, *Rhizoctonia solani* MUCL 9418, and *Pythium ultimum* MUCL 53834 after 144 h of incubation and *Fusarium graminearum* PH-1 after 96 h of incubation at 25°C[a]

| Class | Strain | Identification | BGCs | Antagonistic activity towards: | | | |
|---|---|---|---|---|---|---|---|
| | | | | *Rhizoctonia solani* MUCL 9418 | *Pythium ultimum* MUCL 53834 | *Fusarium oxysporum* MUCL 781 | *Fusarium graminearum* PH-1 |
| Alphaproteobacteria | R-72160 | *Neorhizobium cellulosilyticum* | 10 | E | E | E | E |
| | R-72433 | *Agrobacterium* sp. | 4 | D | D | E | E |
| | R-72456 | *Rhizobium* sp. | 14 | E | E | E | E |
| | R-72066 | *Pararhizobium* sp. | 9 | E | E | E | E |
| | R-73111 | *Phyllobacterium* sp. | 3 | D | D | E | E |
| | R-73088 | *Bradyrhizobium* sp. | 7 | D | D | E | E |
| | R-73074 | *Tardiphaga robiniae* | 5 | E | E | E | E |
| | R-72369 | *Methylopila* sp. | 4 | D | E | E | E |
| | R-72291 | *Caulobacter* sp. | 4 | E | D | E | E |
| | R-71825 | *Roseomonas aerophila* | 8 | E | D | E | E |
| | R-73070 | *Roseomonas hellenica* | 9 | E | D | E | D |
| | R-72501 | *Inquilinus* sp. | 9 | C | C | C | C |
| Betaproteobacteria | R-74557 | *Duganella* sp. | 4 | D | D | E | D |
| | R-73148 | *Duganella* sp. | 6 | D | E | D | D |
| | R-74502 | *Janthinobacterium lividum* | 8 | D | D | D | D |
| | R-72367 | *Achromobacter aestuarii* | 8 | E | E | D | E |
| | R-71975 | *Achromobacter* sp. | 3 | E | E | E | E |
| | R-72016 | *Variovorax* sp. | 10 | C | C | C | C |
| | R-72495 | *Variovorax* sp. | 11 | D | E | D | D |
| | R-72446 | *Variovorax* sp. | 6 | E | E | E | E |
| | R-72060 | *Variovorax* sp. | 6 | D | D | D | D |
| | R-73343 | *Acidovorax* sp. | 4 | E | E | E | E |
| Gammaproteobacteria | R-71998 | *Pseudomonas fluorescens* | 11 | E | D | D | E |
| | R-71838 | *Pseudomonas* sp. | 16 | B | E | B | B |
| | R-72074 | *Pseudomonas fluorescens* | 10 | B | E | B | E |
| | R-72164 | *Pseudomonas atacamensis* | 9 | D | E | B | E |
| | R-72102 | *Pseudomonas paracarnis* | 18 | B | B | B | B |
| | R-72172 | *Pseudomonas paracarnis* | 17 | B | B | B | B |
| | R-71842 | *Pseudomonas orientalis* | 16 | A | A | A | A |
| | R-72008 | *Pseudomonas viridiflava* | 8 | B | B | B | B |
| | R-72210 | *Pseudomonas viridiflava* | 8 | B | B | B | B |
| | R-72599 | *Pseudomonas* sp. | 7 | B | D | E | E |
| | R-72498 | *Pantoea agglomerans* | 7 | D | D | D | D |
| | R-71966 | *Pantoea agglomerans* | 8 | D | B | B | E |
| | R-74235 | *Stenotrophomonas maltophilia* | 4 | D | B | D | D |
| | R-72406 | *Stenotrophomonas* sp. | 2 | D | E | D | D |
| | R-73098 | *Xanthomonas* sp. | 5 | D | D | E | D |
| | R-71986 | *Pseudoxanthomonas* sp. | 3 | D | B | E | E |
| | R-72151 | *Luteibacter* sp. | 11 | C | C | C | C |
| | R-73110 | *Luteibacter* sp. | 7 | C | C | C | C |
| Sphingobacteria | R-72249 | *Pedobacter* sp. | 6 | E | E | E | D |
| | R-72393 | *Pedobacter* sp. | 6 | D | E | E | E |
| | R-74587 | *Pedobacter* sp. | 7 | E | E | E | E |
| | R-72191 | *Pseudosphingobacterium* sp. | 12 | D | D | E | E |
| Flavobacteria | R-74510 | *Flavobacterium* sp. | 7 | E | E | E | E |
| | R-74482 | *Flavobacterium* sp. | 7 | E | D | E | E |
| | R-72247 | *Flavobacterium* sp. | 9 | E | E | E | E |
| Chitinophagia | R-73072 | *Chitinophaga* sp. | 16 | E | D | E | E |
| | R-72609 | *Chitinophaga* sp. | 24 | E | D | E | E |
| | R-72269 | *Chitinophaga* sp. | 13 | E | D | E | E |
| | R-72421 | *Filimonas* sp. | 3 | D | D | D | D |
| Bacilli | R-74298 | *Bacillus cereus* | 11 | B | E | B | B |
| | R-71893 | *Priestia megaterium* | 8 | E | E | E | E |
| | R-74266 | *Brevibacillus* sp. | 11 | NT | NT | NT | E |
| | R-71934 | *Brevibacillus* sp. | 9 | NT | NT | NT | E |
| | R-71971 | *Brevibacillus* sp. | 14 | B | B | B | A |
| | R-74223 | *Brevibacillus borstelensis* | 9 | NT | NT | NT | E |
| Actinomycetia | R-73066 | *Pseudolysinimonas* sp. | 2 | E | D | D | E |
| | R-72356 | *Microbacterium* sp. | 3 | E | E | E | E |
| | R-74611 | *Paenarthrobacter* sp. | 9 | D | E | E | D |
| | R-74106 | *Nocardioides* sp. | 3 | D | D | D | D |
| | R-73050 | *Mycobacterium* sp. | 16 | E | D | E | D |
| | R-71941 | *Tsukamurella* sp. | 14 | D | D | D | D |

[a]The following categories were used to record the effect (examples shown in Fig. 2): category A, bacteria inhibit fungal growth in all directions; category B, bacteria inhibit fungal growth in the vicinity of the bacterial colony; category C, bacteria spread over the plate and inhibit fungal growth upon contact; category D, bacteria inhibit fungal growth upon contact but do not spread; and category E, there is no effect of the bacteria on fungal growth. BGCs were calculated with antiSMASH. NT, not tested.

**FIG 2** Inhibition categories as defined in the dual-culture assay. Representative bacterial treatments for each category are shown against the following phytopathogenic fungi: *Fusarium oxysporum* MUCL 781, *Rhizoctonia solani* MUCL 9418, *Pythium ultimum* MUCL 53834, and *Fusarium graminearum* PH-1. The top row shows controls without bacteria. Category A, bacteria inhibit fungal growth in all directions; category B, bacteria inhibit fungal growth in the vicinity of the bacterial colony; category C, bacteria spread over the plate and inhibit fungal growth upon contact; category D, bacteria inhibit fungal growth upon contact but do not spread.

assay (Fig. S1). The following strains had a significant inhibitory effect on the fungal biomass accumulation after 72 h: *Luteibacter* sp. R-73110, *P. fluorescens* R-71998, *Pseudomonas* sp. R-72599, *Pedobacter* sp. R-72249, *Pedobacter* sp. R-72393, *Brevibacillus* sp. R-71971, and *Pseudomonas* sp. R-71838 (Fig. S1, panel showing green fluorescent protein [GFP] fluorescence corrected for autofluorescence [cGFP]). Among the *Pseudomonas* strains rich in BGCs, *Pseudomonas* sp. R-71838 confirmed its antifungal activity in the detached-leaf assay. Interestingly, *Pedobacter* sp. R-72249 and R-72393, which did not show a particularly high number of BGCs or antagonistic activity in the dual-culture assay, exhibited an inhibitory effect in the experiment. In contrast, while *Pseudomonas fluorescens* R-71998 (11 BGCs) and *Pseudomonas* sp. R-72599 (7 BGCs) showed little or no antifungal activity in the dual-culture assay, they revealed a clear effect in the *in planta* assay.

The three best-performing strains, *Pseudomonas* sp. R-71838, *Brevibacillus* sp. R-71971, and *Pedobacter* sp. R-72393, were selected for further coinoculations (SynComs) in a second detached-leaf assay. Moreover, strains of the collection belonging to these three genera were reevaluated. The compatibility of bacteria for the SynCom experiment was tested on solid medium, and the results indicated that *Brevibacillus* sp. R-71971 exhibited a slight antagonistic effect against *Pedobacter* sp. R-72393 in close proximity, whereas *Pseudomonas* sp. R-71838 inhibited the growth of *Brevibacillus* sp. R-71971 at a distance. The coculture of *Pseudomonas* sp. R-71838 and *Pedobacter* sp. R-72393 did not show any antagonistic effect (Fig. S2).

The results of the SynCom experiment (Fig. 3) showed that while combinations with *Pseudomonas* sp. R-71838 had a significant biocontrol effect ($P < 0.05$ compared to the infected control), this effect did not exceed the biocontrol effect of the single R-71838 treatment (Table S2). Hence, we hypothesize that the *in planta* effect of most of the combinations relies on the effect of R-71838 alone. However, more work is needed to differentiate if a costimulation occurs or if it is the effect of a single dominant strain. More details on the performance of SynComs and individual strains over the course of the 3-day experiment are shown in Fig. S3. Of the other single strains tested, *Pseudomonas* strains R-71842 and

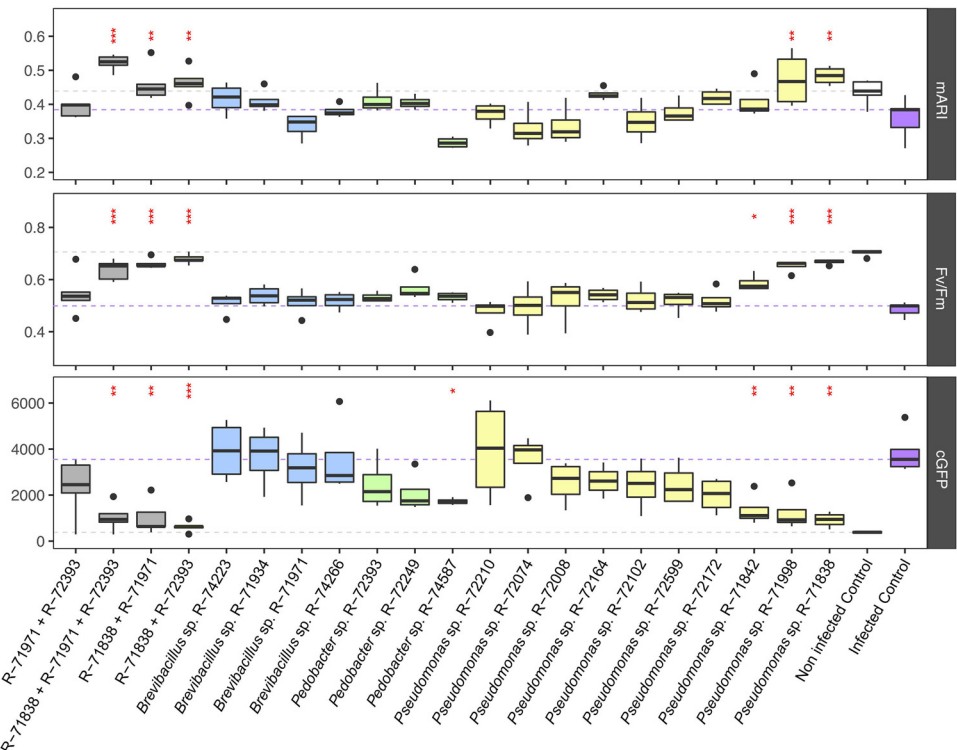

**FIG 3** Boxplots showing the effects of selected bacterial treatments and combinations (SynCom experiment) based on fungal biomass accumulation (GFP fluorescence corrected for autofluorescence [cGFP]), chlorophyll fluorescence (Fv/Fm), and estimation of anthocyanin content (mARI) on detached wheat leaves coinoculated and incubated for a period of 72 h with *Fusarium graminearum* PH-1 ($n = 4$ biological replicates). The noninfected negative control was inoculated with 10 $\mu$L of PBS. Significant differences compared to the infected control are indicated (***, $P < 0.001$; **, $P < 0.05$; *, $P < 0.01$); statistical differences were computed based on the Dunnett test. Groups of the selected bacteria that were retested and combinations are represented by boxes in different colors: *Pseudomonas* strains in yellow, *Pedobacter* strains in green, *Brevibacillus* strains in blue, and combinations in gray.

R-71998 had a significant inhibition effect ($P < 0.05$) on the growth of *F. graminearum* PH-1 (Fig. 3).

Among the 12 strains that strongly reduced the growth of at least three fungi in the dual-culture assay, only 3 strains, *Pseudomonas* sp. R-71838, *Pseudomonas orientalis* R-71842, and *Brevibacillus* sp. R-71971, also did so in the detached-leaf assay or the SynCom experiment. However, only *Pseudomonas* sp. R-71838 exhibited a reproducible antifungal effect in the first detached-leaf assay and the SynCom experiment. Indeed, *Brevibacillus* sp. R-71971 showed a biocontrol activity in the detached-leaf assay but not in the SynCom experiment. Similarly, *Pseudomonas orientalis* R-71842 showed antifungal activity in the SynCom experiment but not in the detached-leaf assay. The reproducible effect of *Pseudomonas* sp. R-71838 highlights its promising potential for field application.

Efficiency of photosystem II, expressed as chlorophyll variable fluorescence/maximum fluorescence (Fv/Fm), values of bacterial treatments inoculated in the absence of *F. graminearum* (1 replicate) were compared against infected and noninfected controls to determine the phytotoxic effect of the bacterial strains on wheat leaflets (Fig. S4). Noninfected controls had an average absolute Fv/Fm value of 0.701 $\pm$ 0.014. The bacteria did not show an evident phytotoxic effect except for the following cases. *Pseudomonas* sp. R-72008 showed an apparent phytotoxic effect with a value of 0.461 (66% of the average value of noninfected controls). Furthermore, two additional *Pseudomonas* strains, R-72210 and R-71842, had a slight phytotoxic effect on detached leaves, with values of 0.609 and 0.629, respectively (corresponding to 87% and 90% of the average values of the noninfected controls). Considering the possible phytotoxic impact of these three strains, we searched for virulence factors encoded by their genomes. They all possess cellulose and pectin-degrading enzymes, particularly multiple pectate lyases, which are well documented as phytopathogenic factors

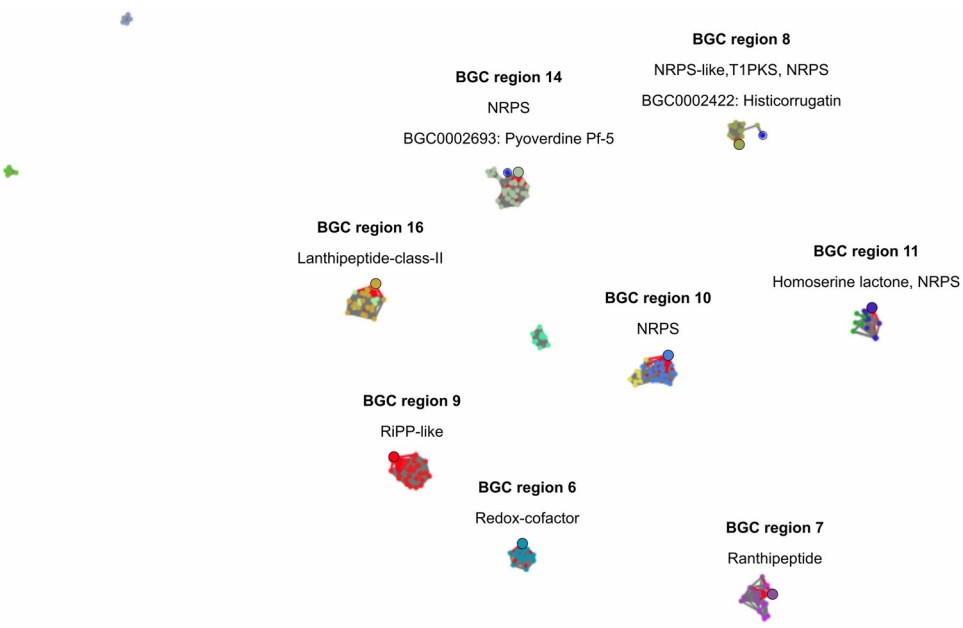

**FIG 4** Similarity network, calculated with BiG-SCAPE on BGCs predicted by cblaster and ClusterBlast (antiSMASH) to match the unique BGCs from *Pseudomonas* sp. R-71838 (BGC regions 6, 7, 8, 9, 10, 11, 14, and 16). Every cluster is indicated as a clan that contains one or more BGC families; different colors in nodes indicate distinct BGC families, and bold circles represent BGCs from *Pseudomonas* sp. R-71838. Known and described matched BGCs contained in the MIBIGs 3.0 database are represented as well.

implicated in the maceration of plant tissue (23, 24), as well as type III secretion systems that are essential as disease effectors (25). Finally, regarding secondary metabolites, antiSMASH predicted *Pseudomonas orientalis* R-71842 to encode pyocyanin, which is a virulence factor for plants, and *Pseudomonas viridiflava* strains R-72210 and R-72008 to possess BGCs encoding cichofactin-like lipopeptides that are related to virulence, motility, and biofilm formation (26). Interestingly, both R-72008 and R-72210 were considered low-BGC strains and thus less interesting for biocontrol purposes.

Overall, our results showed good reproducibility of the effect of *Pseudomonas* sp. R-71838, either alone or in combination with other bacteria, for inhibiting the growth of the fungal pathogen *Fusarium graminearum* PH-1.

**Investigating the genome-encoded features as an alternative for understanding the underlying mechanism for biocontrol and prospection of plant growth promoting rhizobacteria (PGPR) traits.** As *Pseudomonas* sp. R-71838 stood out in all the tests as an effective biocontrol agent against several phytopathogenic fungi, we decided to dig further into the possible mechanisms that make this bacterium unique in our collection. Therefore, the BGC content was compared with that of nine other *Pseudomonas* strains to find specific clusters present exclusively in R-71838 that might be responsible for the remarkable abilities of this strain. Results from the comparison in BiG-SCAPE suggested that eight BGCs are exclusively found in R-71838 (Table S3), six classified as encoding NRPs, one as encoding RiPP, and one as encoding lanthipeptide. Furthermore, the closest BGCs that matched those in R-71838 were retrieved with cblaster and ClusterBlast (antiSMASH) and then clustered with BiG-SCAPE (Fig. 4). BGCs at regions 8 and 14 matched with histicorrugatin and pyoverdine Pf-5, respectively, documented in the MIBIGs database (Table S4). Histicorrugatin is a lipopeptidic siderophore originally described from *Pseudomonas thivervalensis* LMG 21626$^T$ (27), whereas pyoverdine Pf-5 is another siderophore molecule isolated from *Pseudomonas protegens* Pf-5 (28). Both bacteria are well characterized and recognized as promising biological control strains. Even though the other unique BGCs from R-71838 matched those found in other sequenced bacteria (previously deposited in databases), in general, those bacteria contained only one of the BGCs rather than several of them. The fact that *Pseudomonas* sp. R-71838 combines all eight highlights its exceptional biosynthetic potential for producing multiple siderophores and potential antimicrobial molecules.

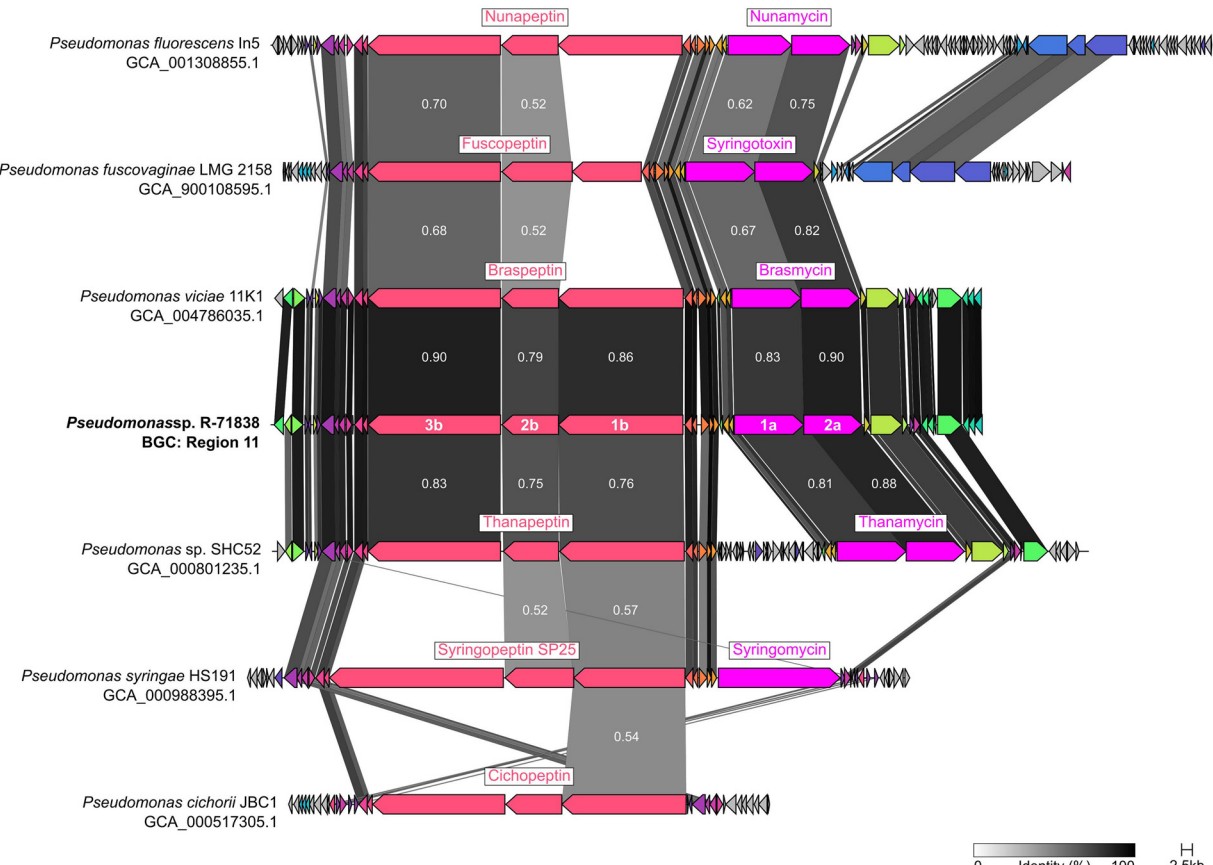

**FIG 5** Gene sequence comparison of BGCs encoding cyclic lipopeptides (CLPs) between *Pseudomonas* sp. R-71838 (BGC region 11, identified by antiSMASH) and other strains with highly similar BGCs. The comparison was computed with the software clinker v0.0.25. Genes 1a and 2a are likely associated with the biosynthesis of a CLP from the mycin family, whereas genes 1b, 2b, and 3b are associated with the peptin family. The similarity is calculated based on amino acid composition and indicated between those proteins with more than 50% similarity (scale, 0 to 1.0).

Of these BGCs not previously described for *Pseudomonas* sp. R-71838, the cluster at region 11 was identified by antiSMASH as having high similarity with the cluster encoding syringomycin, a known secondary metabolite with antifungal activity and virulence factor of the phytopathogen *Pseudomonas syringae* (29–31). To investigate the BGC at region 11 further, we compared it with known clusters sharing homology to syringopeptin/syringomycin (Fig. 5). The results indicate that the BGC encoding braspeptin/brasmycin is the most similar, followed by the thanapeptin/thanamycin cluster. Braspeptin and brasmycin are two cyclic lipopeptides (CLPs) from the peptin and mycin family, respectively; they were described by Zhao et al. (32) for *Pseudomonas* sp. 11K1. Thanapeptin and thanamycin were described by van der Voort et al. (33) for *Pseudomonas* sp. SHC52. All previously mentioned BGCs were initially described as potent biocontrol effectors against diverse phytopathogenic fungi. A structural prediction of the BGC at region 11 and a comparison with braspeptin/brasmycin and thanapeptin/thanamycin clusters are presented in Fig. S5. The comparison highlights that both CLPs from *Pseudomonas* R-71838 represent novel variants from the mycin and peptin family. The mycin-like CLP from *Pseudomonas* sp. R-71838 is 8 amino acids long, and at least one residue is unique compared to brasmycin or thanamycin. Furthermore, the peptin-like CLP is 22 amino acids long and possesses three unique residues compared to braspeptin and thanapeptin.

Besides BGCs present in the genome of *Pseudomonas* sp. R-71838, we also investigated other traits annotated in the genome that might involve additional features for biocontrol, plant growth promotion, and possible virulence factors. Regarding PGPR-related features, we could find the presence of a 1-aminocyclopropane-1-carboxylic acid (ACC) deaminase and a complete pathway for the production of indole acetic acid (IAA). Additionally, a 3-phytase, an enzyme known to enhance phosphate colonization, was annotated

alongside secretion systems II and VI. Type VI secretion systems enhance competitiveness against other prokaryote or eukaryote organisms and are relevant for plant colonization. Despite being related to virulence factors in a few cases, type VI secretion systems are primarily involved in biocontrol activity against other bacteria or fungi (34). Furthermore, no pectin-degrading enzymes were annotated, but a few enzymes involved in cellulose degradation were found; the detection of genes for these traits and the results of *in vitro* assays may denote a natural ability of *Pseudomonas* sp. R-71838 to colonize plants without causing maceration of plant tissue and to exert beneficial effects as plant bioinoculant.

## DISCUSSION

Bioprospection of microorganisms associated with medicinal plants is an opportunity to find interesting candidates with relevant metabolic potential to produce antimicrobials and other secondary metabolites (17–21). In this context, we studied the endophytic bacteria associated with the rhizosphere of the medicinal plant *Alkanna tinctoria*. First, we performed whole-genome sequencing of a selection of 88 bacterial strains representing distinct phylotypes isolated in our previous study (22). We annotated their genomes and quantified the abundance of biosynthetic gene clusters (BGCs). Then, we performed the *in vitro* screening of a subset of bacterial strains by testing strains with different BGC contents. We hypothesized that prioritizing strains with higher counts on BGCs, and thus more biosynthetic potential, would lead to a higher chance of finding biocontrol bacteria with the ability to inhibit the growth of different fungal phytopathogens.

Using genome mining as a tool for revealing interesting bacterial traits has been widely tested in many studies (35–37), and exploring the genomes of bacteria turned out to be an efficient method to reveal biocontrol features in our study, especially those related to the production of antifungal metabolites. While 45 strains did not show a remarkable antifungal activity, 12 strains demonstrated a strong inhibitory effect against at least three pathogenic fungi regardless of the BGC category they belong to (4 high, 4 intermediate, and 4 low in BGCs).

Overall, up to 57% of the bacteria rich in BGCs (4 out of 7 strains), 38% of the bacteria intermediate in BGCs (5 out of 13 strains), and up to 21% of the bacteria low in BGCs (9 out of 43 strains) presented antifungal activity. While we may have missed metabolites of interest by not testing strains with a low number of BGCs, our results confirmed the value of using a genome-mining approach to narrow down the candidate bacteria with the ability to produce antimicrobials. However, strains with a low BGC content may harbor novel mechanisms that are not fully recognized in the genome annotation. To detect and characterize such strains, large-scale antifungal screening remains useful.

Furthermore, we could also recognize some discrepancies between the results of the BGC-based prioritization and the *in vitro* assays. BGCs encode highly diverse sets of secondary metabolites, which are involved in bacterial communication and response against biotic and abiotic stress, including antifungal and antibacterial compounds. Not all predicted BGCs will encode antifungal compounds with efficient activity against the fungi tested in our study. Furthermore, bacterial culture conditions strongly influence the production of secondary metabolites. For example, Basso et al. (38) demonstrated that metabolites and lipids produced by two *Sulfitobacter* strains were strongly determined by growth medium composition, with only 15% of the 175 measured metabolites overlapping across different culture media. In our assay, we observed that bacteria such as *Micromonospora* sp. R-74116 and R-75348 and *Paenibacillus* sp. R-71922 did not grow on potato dextrose agar (PDA) medium. PDA is a very rich medium used for the cultivation of fungi and yeasts. It is widely used for dual-culture assays, along with other rich growth media such as malt agar and yeast dextrose agar, but it is not designed to support the growth of all the diverse bacteria tested in our assay. Moreover, as the nutrient composition of the medium influences the metabolites produced by the bacteria, some antimicrobials might not be produced on PDA medium. On the other hand, some metabolites might be observed only on PDA medium. This is what we suspect for *Pseudomonas* sp. R-71842. This strain shows tiny black granules on its colonies only when grown on PDA and not on R2A, TSA, or 1/10 869 medium (data not shown).

We also noticed an unusual behavior of this strain: it changed the color of the fungi tested. While we could not find literature reports of color change in fungi exposed to bacteria, the effect of *Pseudomonas orientalis* R-71842 on fungal biomass likely involved oxidation or reduction of fungal biomass due to the production of pyocyanin, which was predicted by antiSMASH. Pyocyanin, initially described for *Pseudomonas aeruginosa*, is a secondary metabolite from the phenazine group and is regarded as a virulence factor for plants and animals but also has strong antagonistic effects against fungi and nematodes (39). To overcome the effect of the medium on the metabolic expression of the bacteria, growing bacteria on a wide diversity of media, collecting their metabolites, and exposing the fungi to the extracts can be performed. However, another limitation may occur as some antimicrobials are produced only in cocultivation (40), and growing bacteria in pure culture can miss this information.

Inference of bioactive molecules like antifungals and other secondary metabolites via the prediction of BGCs has limitations as well. Annotation of genomes and prediction of BGCs with bioinformatic tools like antiSMASH might depend on the completeness and contiguity of genomes (41–43). Moreover, the use of short-read sequencing platforms like Illumina, which nowadays is a popular and cost-effective technology, tends to produce fragmented genome assemblies. Such fragmentation in assemblies is often due to repetitive sequences or regions with high GC content in the genomes. Therefore, the prediction of BGCs based on these highly fragmented genomes might produce an artificially inflated count of BGCs. For this reason, we selected some representative genomes with suspiciously high counts of BGCs to perform a complementary nanopore sequencing, with the objective of obtaining contiguous and fully represented genomes. With this strategy, we not only improved the accuracy of the prediction of BGC counts for some potentially interesting bacteria like *Pseudomonas* sp. R-71838 but also ensured a full representation of such BGCs that can be connected to their encoded metabolites via prediction or comparison with described BGCs.

Despite the limitation of genomic and *in vitro* studies, the majority of the bacteria selected based on the potential to encode abundant BGCs showed strong antifungal activity in the dual-culture assay, making them promising candidates in the upscaling of biocontrol metabolite production. Future research concerning a dose-response effect of those metabolites can provide insight into the potential to (i) further reduce fungal growth, (ii) inhibit fungal sporulation, or (iii) completely kill the fungus. Similarly, most of the strains with no antifungal activity were shown to have a small amount of BGCs.

We also observed differences between the dual-culture assay and the detached-leaf experiment. Inconsistent results in the application of biocontrol bacteria are widely known, as the outcome depends strongly on the successful establishment and colonization of the microorganisms (44). For example, Besset-Manzoni et al. (45) highlighted that bacteria showing no biocontrol activity *in vitro* were efficient against *F. graminearum in planta*. Among the bacteria without *in vitro* activity in their study, *Microbacterium*, *Arthrobacter*, and *Variovorax*, which are not known in the literature for antifungal properties, were able to protect the plants. Moreover, the activity of bacteria *in vitro* might not be reproducible *in planta*. In their study, Bacilio et al. (46) tried to alleviate salt stress in pepper plants using *Pseudomonas stutzeri* strain TREC and compared the effect of bacterial inocula in different systems. The effect of the bacteria was reduced under greenhouse conditions and not conclusive under field conditions compared to the *in vitro* study. This lack of correlation is linked to differences in environmental conditions between *in vitro* and *in planta* experiments (47, 48).

Our genomic analysis and biocontrol experiments revealed *Pseudomonas* sp. R-71838 as a potential biocontrol agent. This strain stood out by its reproducible effect during the *in planta* tests and represents a valuable resource as a potential biocontrol inoculant. The large number of BGCs (16) in its genome reflects its superb biosynthetic potential, with at least one of these responsible for the production of novel cyclic lipopeptides and siderophores that merit further study. Such BGCs are likely involved in the consistent ability of this bacterium to inhibit the proliferation of fungi in our *in vitro* assays. Annotation of features like type VI secretion systems involved in biocontrol enhancement and additional biostimulant features corroborates the value of this bacterium as a promising bioinoculant

for plant protection and growth enhancement. Besides the direct biocontrol effect of R-71838, likely due to the production of antifungal metabolites, we can also expect that this bacterium might be able to prime plant immune defenses through the induction of the secondary metabolism, as described by Varela Alonso et al. (49).

Finally, as synthetic communities have received increased interest in past years due to their enhanced effects as bioinoculants compared to single strains and because of their wide applicability (50), we tested combinations of bacteria to inhibit fungal growth. Although our research could not highlight a cumulative consortium effect, the added value of the SynComs should be evaluated in a more complex field setting with an innate microbiome to test its persistence in the environment. This highlights the need for more extensive research on SynComs to discover efficient cooperation and assess their performance.

Our study demonstrated the advantage of exploring a taxonomically diverse set of bacteria and the suitability of genomic information to guide the bioprospection of biocontrol bacteria. Several underexplored genera had interesting genomic potential, and many strains were able to produce effective antifungals in the dual-culture assay. In view of these benefits, our collection of genomes from endophytic bacteria associated with *Alkanna tinctoria* could be an interesting resource for future metabolomic studies.

## MATERIALS AND METHODS

**Bacterial collection.** A selection of 88 bacterial strains representing distinct phylotypes isolated in our previous study (22) was used for genome sequencing (Table S1). Culture stocks maintained at −80℃ were retrieved and regrown at 28℃ in R2A medium.

**Genomic DNA extraction.** Strains were grown on R2A medium for 48 h, and genomic DNA was extracted from fresh biomass with a Maxwell 16 tissue DNA purification kit in a Maxwell 16 instrument (Promega), according to the manufacturer's instructions. DNA extracts were treated with RNase (2 mg mL$^{-1}$) and incubated at 37℃ for 1 h. DNA integrity was assessed in an agarose gel (0.7%) by electrophoresis; DNA quantification was performed using a Quanti-Fluor ONE double-stranded DNA (dsDNA) system and a Quantus fluorometer (Promega).

**Genome sequencing and annotation of functional traits.** Genomes were sequenced on the Illumina NovaSeq 6000 platform (paired-end 150 bp reads) at the Oxford Genomics Center (University of Oxford, United Kingdom). To improve the quality and contiguity of assemblies for some strains of interest, genomic DNAs were also sequenced in-house with the Oxford Nanopore Technologies MinION platform, using R9.4 flow cells and the ligation library preparation kit with barcoded samples. DNA library preparation and sequencing were done according to the manufacturer's protocol, version NBE_9121_v109_revF_19Jan2021.

Reads generated with Illumina NovaSeq 6000 were trimmed using Trimmomatic v0.36 (51). Meanwhile, files generated by the Nanopore sequencer were base called, demultiplexed, trimmed, and quality checked using Guppy v5.0.7.

Genomes sequenced with Illumina data only were assembled with Shovill v1.0 (https://github.com/tseemann/shovill). Hybrid genome assemblies for selected strains were performed with the preprocessed data from Oxford Nanopore Technologies and Illumina using Unicycler v0.4.8 (52). Genome-based taxonomic classification was performed in GTDB-Tk v2.0.0, database version R06-RS202 (53).

Genomes were deposited and annotated in the online server Integrated Microbial Genomes & Microbiomes system (IMG/M) (54). The full list of accession numbers is provided in Table S1 in the column with the head IMG ID. Completeness and quality of genomes were assessed by CheckM v1.1.3 (55), and average nucleotide identity (ANI) with FastANI v1.33 (56) was used for genome dereplication.

antiSMASH v6.1, with relaxed prediction settings (57), and BiG-SCAPE v1.15 (58) were used to investigate the distribution of biosynthetic gene clusters (BGCs) in bacterial genomes for the complete collection.

Furthermore, cblaster (59) at the CAGECAT server (https://cagecat.bioinformatics.nl) and results for *Pseudomonas* sp. R-71838 from the ClusterBlast module (antiSMASH) were used to retrieve the genomes of bacteria containing the closest matches for specific or unique BGCs in *Pseudomonas* sp. R-71838. All BGCs were extracted and reclustered in BiG-SCAPE v1.15 (58) using the MIBiG 3.0 database for known and described BGCs.

**Dual-culture assay.** A subselection of 63 bacterial strains was chosen for further phenotypic tests to look for antagonistic activity against fungi. This selection included 7 strains rich in BGCs (16 or more BGCs), 13 strains randomly chosen out of 19 with an intermediate number of BGCs (between 10 and 15 BGCs), and 43 strains randomly selected out of 61 that were poor in BGCs (fewer than 10 BGCs).

Bacteria were tested for activity against four phytopathogenic fungi known to infect various host crops and strongly impact their yields: *Rhizoctonia solani* MUCL 9418, *Pythium ultimum* MUCL 53834, *Fusarium oxysporum* MUCL 781, and *Fusarium graminearum* PH-1. The fungi tested were grown on potato dextrose agar (PDA) at 25℃ for 1 week. The selected bacteria were grown in 10 mL of R2B liquid medium for 24 h to reach an optical density (OD) of approximately 0.9. For each strain on a PDA plate, 20 μL of the liquid culture was spotted twice, once in each half of the plate. The plates were then incubated at 28℃ for 72 h. After 72 h of incubation, a plug with a diameter of 0.5 cm was cut out from the mycelia of the fungi. The plug was then placed upside down (mycelium side in contact with the agar) at the center of the PDA plate at a distance of

about 2.5 cm from the bacteria. Control plates without bacteria were also prepared for each fungal pathogen: plates with only fungal plugs and plates with fungal plugs and the addition of two spots of 20 $\mu$L of the sterile R2B liquid medium. The plates were incubated at 25℃ and checked every day until the fungi covered the control plate. Two technical replicates were prepared per bacterium and fungus tested.

**Detached-leaf assay.** A GFP-transformed *F. graminearum* PH-1 strain was used in the detached-leaf experiment (60). *F. graminearum* PH-1 was grown on potato dextrose agar (PDA) for 7 days at 21℃ under a regime of 12 h of darkness and 12 h of combined UVA and UVC light to induce sporulation. Spores were harvested by adding approximately 20 mL of phosphate-buffered saline (PBS; pH 7.2 to 7.4) containing 0.01% of Tween 80 to the petri dishes and smeared on the mycelium with a Drigalski spatula. Spores were separated from the mycelium by filtering the suspension through a sterile Miracloth filter and were diluted to a concentration of $1 \times 10^6$ spores·mL$^{-1}$. Bacterial isolates were grown overnight at 28℃ in R2A and washed in sterile PBS; bacterial biomass was resuspended in PBS and adjusted to an OD of 0.5.

Wheat plants (cv. Tybalt) were grown in potting soil for 10 days (temperature, 21℃; relative humidity [RH], 40%; 16 h:8 h light/dark), after which a bio-assay was performed on detached leaves (approximately 6 cm). The leaves (leaf tops) were placed on their abaxial surface in rectangular petri dishes (VWR; 734-2977) containing 0.5% (wt/vol) water agar supplemented with 40 mg·L$^{-1}$ of benzimidazole. The center of the leaves was wounded with a scalpel by gently scraping the epidermal layer of the leaves. Wheat leaflets were inoculated with 5 $\mu$L of *F. graminearum* PH-1 spore suspension ($1 \times 10^6$ spores·mL$^{-1}$) and coinoculated with 5 $\mu$L of bacterial suspension. An infected positive control was included in the experiment, which was coinoculated with 5 $\mu$L of sterile PBS instead of the bacterial suspension. Additionally, a noninfected negative control was included, which was inoculated with 10 $\mu$L of PBS. All treatments and controls were included in four biological replicates. Finally, the (possible) phytotoxic effect of the bacterial strains on wheat leaflets was evaluated by coinoculating the bacterial suspensions with PBS but not fungal spores. The disease progression in the detached leaves was monitored through multispectral imaging after 0, 24, 48, and 72 h (details provided below).

The three best-performing strains were chosen from an initial selection of strains to compose synthetic microbial communities (SynCom). To check for mutual antagonism, they were spotted as combinations of two strains on R2A plates and grown for 96 h. Finally, the detached-leaf assay was repeated following the previous protocol, with the difference that the bacterial inoculum was prepared by combining equal volumes of individual bacterial cultures (OD = 1) and adjusting to a final OD of 0.5 for the SynCom treatments (total inoculum, 5 $\mu$L). Additional bacterial strains belonging to the same species as the best-performing strains were also tested as individual treatments to provide a comparative framework, and inoculum was prepared as described previously for the first experiment.

An automated platform (CropReporter; Phenovation, Wageningen, The Netherlands) was used to evaluate the detached-leaf assay, allowing real-time monitoring of several physiological traits based on specific absorption, emission, and reflection patterns. Using a charge-coupled-device (CCD) 6Mp-16 bit camera system, spectral parameters, including red green blue (RGB) values, chlorophyll fluorescence (Fv/Fm), chlorophyll index (ChlIdx), and GFP fluorescence, could be monitored at a high spatial and temporal resolution.

Fungal biomass accumulation was monitored through the GFP signal of *F. graminearum* PH-1, corrected for autofluorescence due to leaf senescence (cGFP). The efficiency of photosystem II in a dark-adapted state was measured as the chlorophyll fluorescence (Fv/Fm). It was used as a proxy for leaf health and concomitant stress levels (61). The chlorophyll index (ChlIdx), a proxy for the chlorophyll content of leaves, was used to measure leaf greenness (62). Finally, the modified anthocyanin reflectance index (mARI) was used as a measure for anthocyanin accumulation, which can be associated with oxidative burst and the plant defense system (63).

**Investigation of BGCs expected to encode secondary metabolites linked to antifungal activity in *in vitro* assays.** BGC region 11 from *Pseudomonas* sp. R-71838, a cluster suspected to encode secondary metabolites with antifungal activity, was investigated in more detail. To do so, we compared this BGC to others like the syringopeptin/syringomycin cluster from *Pseudomonas syringae* HS191 (GenBank Assembly accession no. GCA_000988395.1). In the comparison, we included the clusters encoding cichopeptin from *Pseudomonas cichorii* JBC1 (GCA_000517305.1), fuscopeptin/syringotoxin from *Pseudomonas fuscovaginae* LMG 2158 (GCA_900108595.1), nunapeptin/nunamycin from *Pseudomonas fluorescens* In5 (GCA_001308855.1), braspeptin/brasmycin from *Pseudomonas viciae* 11K1 (GCA_004786035.1), and thanapeptin/thanamycin from *Pseudomonas* sp. SHC52 (GCA_000801235.1). All clusters were predicted by antiSMASH v6.1 (57) from original genomes retrieved at the NCBI database. BGCs were compared using the software clinker v0.0.25 (64).

**Data processing and statistical analysis.** For the statistical analysis of the multispectral data, the data were grouped per time point. Statistical differences between the infected control and biocontrol treatments, or between the SynComs and members thereof, were evaluated in IBM SPSS statistics (v25.0; Armonk, NY, USA) by means of analysis of variance (ANOVA) and *post hoc* Dunnett test after verifying normality and homoscedasticity.

**Data availability.** Genomes were deposited and annotated in the online server Integrated Microbial Genomes & Microbiomes system (IMG/M [https://img.jgi.doe.gov/]). The full list of accession numbers is provided in Table S1 in the column with the head IMG ID.

## SUPPLEMENTAL MATERIAL

Supplemental material is available online only.

**SUPPLEMENTAL FILE 1**, XLSX file, 0.1 MB.

**SUPPLEMENTAL FILE 2**, DOCX file, 3.1 MB.

## ACKNOWLEDGMENTS

We thank the Oxford Genomics Centre at the Wellcome Centre for Human Genetics (funded by Wellcome Trust grant reference 203141/Z/16/Z) for the generation and initial processing of the sequencing data. We acknowledge Stéphane Declerck and the BCCM/MUCL culture collection for providing some of the fungal strains used in this study.

We declare that we have no known competing financial interests or personal relationships that could have appeared to influence the work reported in this paper.

This research was supported by the Europe Union's Horizon 2020 research and innovation program under Marie Sklodowska-Curie grant agreement no. 721635 (project Micrometabolite) and by the Special Research Fund BOF of Ghent University (grants 01IT0720 and 01IT0121). This study was carried out using infrastructure funded by EMBRC Belgium—FWO international research infrastructure I001621N. The computational resources (Stevin Supercomputer Infrastructure) and services used in this work were provided by the Flemish Supercomputer Centre (VSC), funded by Ghent University, FWO, and the Flemish Government, department EWI. We acknowledge the Hercules initiative for the multispectral imaging platform, grant AUGE/15/17.

Henry D. Naranjo: Conceptualization, Investigation, Methodology, Formal Analysis, Software, and Writing – Original Draft. Angélique Rat: Conceptualization, Investigation, Methodology, Formal Analysis, and Writing – Original Draft. Noémie De Zutter: Investigation, Methodology, Formal Analysis, Software, and Writing – Original Draft. Liesbeth Lebbe: Investigation and Validation. Emmelie De Ridder: Investigation and Validation. Kris Audenaert: Supervision, Funding Acquisition, and Writing – Review & Editing. Anne Willems: Supervision, Funding Acquisition, Project Administration, and Writing – Review & Editing.

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
