## [Reviewer comments · Microbiology Spectrum]

Microbiology Spectrum

Uncovering genomic features and biosynthetic gene clusters in endophytic bacteria from roots of the medicinal plant *Alkanna tinctoria* Tausch as a strategy to identify novel biocontrol bacteria

Henry Naranjo, Angélique Rat, Noémie De Zutter, Liesbeth Lebbe, Emmelie De Ridder, Kris Audenaert, and Anne Willems

Corresponding Author(s): Anne Willems, Universiteit Gent

Review Timeline:

Submission Date:	February 17, 2023
Editorial Decision:	March 20, 2023
Revision Received:	June 23, 2023
Accepted:	June 25, 2023

Editor: Frédérique Reverchon

Reviewer(s): The reviewers have opted to remain anonymous.

Transaction Report:

DOI: <https://doi.org/10.1128/spectrum.00747-23>

March 20, 2023

Dr. Anne Willems
Ghent University
Belgium

Re: Spectrum00747-23 (Uncovering genomic features and biosynthetic gene clusters in endophytic bacteria from roots of the medicinal plant *Alkanna tinctoria* Tausch as a strategy to identify novel biocontrol bacteria)

Dear Dr. Anne Willems:

I have now received the comments of two independent reviewers on your manuscript. One reviewer was very encouraging whilst the other expressed strong criticism towards your study. I am prepared to evaluate a revised manuscript provided you can address those comments.

As you will see, both reviewers expressed the need to understand the selection of the host species for bacterial isolation. Methods and reported results also need to be checked for consistency. Conclusions should also be tuned down to take into account that additional experimental assays are needed to ensure that genomic analysis provided a useful and efficient approximation to select for antifungal strains. As one of the reviewer mentioned, what about the non-selected strains? Are you sure they could not antagonize *Fusarium*?

Link Not Available

Sincerely,

Frédérique Reverchon

Journals Department
Reviewer comments:

Reviewer #1 (Comments for the Author):

The manuscript describes the characterization of several bacterial species with antifungal properties. Based on knowledge of the genome of a collection of bacterial strains isolated from the medicinal plant *A. tinctoria*, the authors characterized their properties as biocontrol agents. Some of the tested strains had a marked inhibitory effect on *Fusarium gramineum*. These strains belong to *Pseudomonas*, *Pedobacter*, and *Brevibacillus* species and, either singly or in combination, maintain their antagonistic properties against various fungal plant pathogens. From the analysis of 80 strains, one of potential interest for biocontrol experiments emerged.

The authors give a good introduction to the area of discovery of PGPR and biocontrol bacteria. They reckon that genomic information would facilitate the search for this kind of bacteria. However, the relevant finding was that only one bacterial candidate with the best performance in biocontrol was obtained after a broad screening. So then, more than genomic information is needed to predict antimicrobial activities.

Specific questions:

- What is the relevance, in any, that the strains come from *A. tinctoria*.
- Among the distinct phenotypes displayed in the inhibition assay, what is the most important for biocontrol purposes? By seeing the plates, it could be supposed that the A phenotype is the best, but this is the less common at once. Noteworthy, there are no bacteria able to kill fungi in the assay.
- What are the genomic differences between *Pseudomonas* sp R72008, the worst strain in the Fv/Fm assay (supp. Fig.), and *Pseudomonas* sp R71838 the best strain?
- There are genomic differences in the *Pseudomonas* collection regarding R 71838 (Table 2). R71838 has more diverse BGCs than the other isolates. These are known in other bacteria. In Table 2, where does the most similar cluster come from? The similarity was calculated with respect to what cluster? Is it expressing the percentage of nucleotides or amino acids? It seems very low.
- In this sense, R71838 is unique in the *Pseudomonas* world concerning the BGCs content. Are there other *Pseudomonas* in databases with these features?
- In supp. Fig. 3, what is the uninfected control?
- Combinations of strains do not seem to increase the protection against fungi. The main contribution in the assays comes from a single strain. For example, the efficiency of R71383 combined with *Pedobacter* strain R72393 is not synergistic, only marginal with the single strains.

Reviewer #2 (Comments for the Author):

Henry D. Naranjo et al report a genomic and phenotypic study to investigate biocontrol properties of endophytic bacteria isolated from *Alkanna tinctoria*. Authors performed a fair amount of work including, Isolation, whole genome sequencing, in vitro and in vivo testing of strains against a subset of fungal pathogens. The manuscript is well written, most experiments are well designed and most of their conclusions are supported by the results obtained. However, I have several concerns with the way the experiments were performed and the corresponding interpretation. For instance, authors claim that they performed a strain prioritization step prior performing in vitro testing. The results they show have several inconsistencies and, overall, they failed to demonstrate that their criteria lead to an effective process of strain prioritization. For instance, they did not demonstrate that strains not picked do not have any antifungal activity. On the other hand, authors showed that the selected strains have the potential to produce antifungal compounds but did not show any experimental (or in silico) evidence supporting that such strains are valuable or different to many other strains obtained in other similar studies. In other words, I think that the work presented requires much more experimental work (e.g. identification of potential antifungal molecules, screening of hundreds of different strains to demonstrate an effective process of strain prioritization) in order to demonstrate significance to the scientific community.

Major comments:

- I consider that the Introduction section is too long. There are some arguments that can be easily avoided or rewritten to build a compelling argument around the topic. For instance, paragraphs 2-5 could be replaced with a single paragraph stating that microbes are a feasible alternative to treat plants, providing with detail some examples. Authors also provide, in very small paragraphs, several applications of microbes in agriculture. There is no information regarding the rational (ecological or practical) of isolating strains from *A. tinctoria*. Overall, I think the introduction section is not well organized and makes difficult to understand the significance and originality of the study.
- The organization of the Results/Discussion section should be modified. There are a few things that are missing or not covered with sufficiency. For example, in the methods section, the authors include a section of "Evaluation of disease progression..." that is not identified in the main Results section.
- Authors claim that they performed a strain-prioritization process for features related to plant stimulation and biocontrol based on genomic information. It seems difficult for me to believe that in silico testing (genomic information -BGC structure) can lead to

perform a bioactivity-based prioritization exercise. In fact, the data provided by the authors seem weak on this regard. My guess is that, following the procedure described, authors probably lost valuable candidates that encoded uncharacterized BGCs with relevant bioactivities (antifungal or growth promoting).

First, the prioritization (or strain selection) was performed using known genetic signatures or enzymes involved in the metabolism of known molecules (IAA, ACC). This experimental approach affects the originality of the results and overall significance of the study.

Second, from figure 1, I see that there are a few isolates that have prominent amount of BGCs that were not selected (and a few selected strains that did not show relevant genetic features or prominent amount of BGCs). Is there any additional information why authors selected such strains? I suggest including in the methodology section the entire selection criteria. Otherwise, the prioritization criteria seem inconsistent. For example, why did R72151 was not picked and R72406 was picked. According to the selection criteria provided, it should have been the other way around. Strains without prominent antifungal/growth promoting activity are expected - please include this in the results and discussion section.

- What was the criteria used to perform double sequencing strategy. Supplementary table 2 lists genome assemblies that are very heterogeneous. Is there any particular reason why to include in the analysis genomes with more than 100 contigs. BGC analysis of highly fragmented genomes yields unreliable/redundant results. Given the nature of the study, I would like to see the AAI value for each isolate.

- Line 405. I don't think this conclusion is supported by the data provided by the authors. As mentioned before, there are a few experiments that need to be performed to assure that the genomic analysis provided lead the authors to identify strains with biocontrol activity against the pathogens tested. If authors want to claim that they performed an efficient strain prioritization, they should demonstrate it by testing that strains not selected do not have the relevant bioactivity (using co-culture and crude extract assays).

Minor comments

- In the abstract section, authors mention that plant -growth promoting activity was tested as part of the study, however there are no results on this regard in the manuscript. Please modify the sentence.

- Line 187. Please provide a justification for the selection of the fungal strains for the co-culture experiments.

- Line 193. Please provide a sentence describing the overall results obtained in the screening. How many strains showed strong activity? how many strains showed weak activity? and how many strains did not show the expected activity? Also, include this results in the discussion section.

- Line 216 and again in line 236. Please remove such sentence unless it is properly described in the strain prioritization process.

- Line 242. Specify how many strains were selected and why?

- Please expand the discussion section by adding arguments regarding the media used to trigger secretion of natural products.

What other culture media has been previously used in similar studies? Authors should include in the discussion section that this type of problem can be easily solved by using crude extracts for bioactivity testing.

- I recommend authors to perform NRPS analysis (supplementary figure 5) with complementary tools, such as adenylation domain specificity (NRPSpredictor2) or other tools that provide more detailed information.

- I recommend authors to perform a comparative study for the complete BGCs detected using BigSlice. This will provide an indication of how many different BGCs (based on GCFs) are within the strains selected and how many are potentially novel compounds. In addition, I think Table 2 should be removed as the information provided from AntiSmash pipeline (% similarity) is sometimes not precise. I strongly suggest substituting Table 2 for a more relevant comparative analysis such as BigSlice comparative network using only complete BGCs.

- Did authors perform the co-culture assay with other isolates that did not pass their prioritization stage. This might be relevant to report in case it was performed. Given what authors describe in the manuscript, I doubt that all the excluded strains do not encode for any antifungal metabolite.

Staff Comments:

Preparing Revision Guidelines

- Point-by-point responses to the issues raised by the reviewers in a file named "Response to Reviewers," NOT IN YOUR COVER LETTER.
- Upload a compare copy of the manuscript (without figures) as a "Marked-Up Manuscript" file.

- Each figure must be uploaded as a separate file, and any multipanel figures must be assembled into one file.
- Manuscript: A .DOC version of the revised manuscript
- Figures: Editable, high-resolution, individual figure files are required at revision, TIFF or EPS files are preferred

Please return the manuscript within 60 days; if you cannot complete the modification within this time period, please contact me. If you do not wish to modify the manuscript and prefer to submit it to another journal, please notify me of your decision immediately so that the manuscript may be formally withdrawn from consideration by Microbiology Spectrum.

Rebuttal Letter to manuscript ID: Spectrum00747-23

“Uncovering genomic features and biosynthetic gene clusters in endophytic bacteria from roots of the medicinal plant *Alkanna tinctoria* Tausch as a strategy to identify novel biocontrol bacteria”

Dr. Frédérique Reverchon
Editor

June 19th, 2023

Dear Dr. Frédérique Reverchon and dear reviewers,

We would like to thank you for all the suggestions and comments. We are grateful for the extra time which allowed us to test some more strains in the dual culture assay so we could better address the critical comments raised. This allowed us to revise and improve our manuscript thoroughly.

We need to explain that an extra co-author, Emmelie De Ridder, was added on the revised manuscript. Emmelie is a PhD student who has performed the additional dual culture assays in the UGent lab, since Henry Naranjo and Angélique Rat have both moved to positions elsewhere after completing their PhD studies at UGent and could not themselves perform this work.

Thank you for reconsidering our revised manuscript for publication in Microbiology Spectrum.

Below we reply to all the points raised by the reviewers and the editor.

Kind regards,

Anne Willems, on behalf of all co-authors

Reviewer#1

Dear reviewer 1,

We appreciate the fast and thorough feedback on our manuscript and thank you for the comments that have allowed us to improve the article. Below we provide a reply to all the points one by one:

- What is the relevance, in any, that the strains come from *A. tinctoria*.

It has been suggested that the close relationship between plants and endophytes might result in the production of a high number and a wide diversity of molecules with bioactive properties. This biological relationship endophyte-plant as a source of bioactive molecules is even more interesting in the case of plants already known for medicinal purposes. Moreover, several studies have shown the antimicrobial properties of bacteria isolated from medicinal plants. For these reasons, we wanted to explore the biological properties of bacteria isolated from the medicinal plant, *Alkanna tinctoria*. This has now been clarified in the main text (lines 106-113).

- Among the distinct phenotypes displayed in the inhibition assay, what is the most important for biocontrol purposes? By seeing the plates, it could be supposed that the A phenotype is the best, but this is the less common at once. Noteworthy, there are no bacteria able to kill fungi in the assay.

Many criteria are interesting for antifungal biocontrol: killing the fungi, growth inhibition, and sporulation inhibition. In our work, we chose to focus on metabolites totally or partially inhibiting fungal growth. We hypothesize that many metabolites responsible for the B phenotype have the potential to kill the fungi if produced and inoculated in high concentrations. Future metabolite purification and batch production should be performed in order to isolate and assess the effect of such metabolites at different concentrations (i.e. dose-response experiments). We added a brief statement on this matter to the discussion (lines 399-405).

- There are genomic differences in the *Pseudomonas* collection regarding R-71838 (Table 2). R71838 has more diverse BGCs than the other isolates. These are known in other bacteria. In Table 2, where does the most similar cluster come from? The similarity was calculated with respect to what cluster? Is it expressing the percentage of nucleotides or amino acids? It seems very low. Is it expressing the percentage of nucleotides or amino acids?

In Table 2 (now Table S3), we displayed the raw information predicted by antiSMASH, a specialised software to study and analyse Biosynthetic Gene Clusters (BGCs). These results come from the comparison against a reduced database of genomes previously processed by antiSMASH; it does not contain all sequenced bacteria (a global database). To cover this gap, we used cblaster, which compares our predicted BGCs against those contained in the NCBI database. For more details, we included Figure 4, Table S4 and added some clarification in the results section (lines 278-289).

- It seems very low. Is it expressing the percentage of nucleotides or amino acids? In this sense, R71838 is unique in the *Pseudomonas* world concerning the BGCs content.

It is calculated based on amino acid composition, and indeed, it is low for most of the BGCs; this indicates that they are rare or not well represented in the antiSMASH database. Our extended analysis with cblaster found matches, but these mainly represent less well-described BGCs (probably producing novel and interesting compounds). The novelty of this bacterial strain relies on the fact that it possesses a very extended range of BGCs that each can only be found in a rather diverse group of *Pseudomonas* spp. Furthermore, many of these BGCs might be less well-described BGCs encoding for interesting molecules.

- In supp. Fig. 3, what is the uninfected control?

The non-infected negative control was inoculated with 10 μ L PBS. The figure and its associated legend were updated accordingly.

- Combinations of strains do not seem to increase the protection against fungi. The main contribution in the assays comes from a single strain. For example, the efficiency of R71383

combined with Pedobacter strain R72393 is not synergistic, only marginal with the single strains.

This is indeed a valid point. In the manuscript and in Figure 3, we only included a pairwise comparison of the efficacy of the treatments against the infected control to show potential biocontrol activity. To address the point raised, we did an additional statistical analysis to compare the effect of the combined strains to the effect of the single strains (or other combinations of these single strains). We have adapted this section in the manuscript (lines 225-230) and added the pairwise comparison as per ANOVA and post-hoc Dunnett-test in Table S2. Additionally, we elaborated on the added value of SynComs in the discussion section (lines 435-441).

Reviewer#2

Dear reviewer 2,

Major comments:

- I consider that the Introduction section is too long. There are some arguments that can be easily avoided or rewritten to build a compelling argument around the topic. For instance, paragraphs 2-5 could be replaced with a single paragraph stating that microbes are a feasible alternative to treat plants, providing with detail some examples. Authors also provide, in very small paragraphs, several applications of microbes in agriculture. There is no information regarding the rationale (ecological or practical) of isolating strains from *A.tinctoria*. Overall, I think the introduction section is not well organized and makes difficult to understand the significance and originality of the study.

We agree that the introduction section was too extensive and have updated the manuscript. Indeed, the rationale for the use of *A. tinctoria* in the isolation of bacterial strains was not well-explained. Addressing this point raised by both reviewers, we provided an explanation of the relevance of using bacteria isolated from a medicinal plant (lines 106-113). We hope the introduction is now better organized, while still providing the information needed for the reader.

- The organization of the Results/Discussion section should be modified. There are a few things that are missing or not covered with sufficiency. For example, in the methods section, the authors include a section of "Evaluation of disease progression..." that is not identified in the main Results section.

We have modified the text by separating the Results and Discussion sections to fit the formatting requirements of the journal and to improve the readability of our manuscript. We apologise for the discrepancy between the headings in the material and methods and the results section. We have merged several sections for clarity. In "Evaluation of disease progression..." we explained the technology we used to evaluate the Detached leaf assay. However, making this a separate section was not appropriate; both sections were merged in our updated manuscript.

- Authors claim that they performed a strain-prioritization process for features related to plant stimulation and biocontrol based on genomic information. It seems difficult for me to believe that in silico testing (genomic information -BGC structure) can lead to perform a bioactivity-based prioritization exercise. In fact, the data provided by the authors seem weak on this regard. My guess is that, following the procedure described, authors probably lost valuable candidates that encoded uncharacterized BGCs with relevant bioactivities (antifungal or growth promoting).

We appreciate this comment and critically reviewed our data and how we describe our findings. We agree on the need to include more strains and have now performed dual culture assays on additional strains and included these results. We have also reworked our Results and Discussion

sections to make a more clear argument. Out of the 88 strains we sequenced, we chose 63 strains (up from the previous 55, Table 1) for prioritisation in the *in vitro* tests. The selection was based on the various BCGs categories (strains with high, intermediate and poor in BGCs), now we have tested at least 70% of the strains in every category, and a detailed description can be found in lines 163-170. We confirmed that selected strains, particularly those with a high number of BGCs (16 or more), were the best performers in our *in vitro* studies, while most of the ones poor in BGCs (less than 10) did not have an antifungal effect. There were a few exceptions, as we explain in lines 353-371. We do not now propose the use of genomic analysis as a unique strategy to select bacteria with bio-active potential as; indeed, strains with novel BGCs might be missed. Still, we encourage the use of genomic tools to speed up the study of a wide diversity of bacteria that necessary need to be confirmed with *in vitro/in vivo* studies.

First, the prioritization (or strain selection) was performed using known genetic signatures or enzymes involved in the metabolism of known molecules (IAA, ACC). This experimental approach affects the originality of the results and overall significance of the study.

We apologise for the misunderstanding. Our selection was based on the BGCs richness, not other features. We provided PGPR information as additional data of interest for biostimulant/biocontrol microbial inoculants, as explained in lines 154-159.

Second, from figure 1, I see that there are a few isolates that have prominent amount of BGCs that were not selected (and a few selected strains that did not show relevant genetic features or prominent amount of BGCs). Is there any additional information why authors selected such strains? I suggest including in the methodology section the entire selection criteria. Otherwise, the prioritization criteria seem inconsistent. For example, why did R72151 was not picked and R72406 was picked. According to the selection criteria provided, it should have been the other way around. Strains without prominent antifungal/growth promoting activity are expected - please include this in the results and discussion section.

Our initial selection was based on various content of BCGs, and aimed to represent the taxonomic diversity of the strains, and thus not all the strains were included. Considering the previous comments above, we realised we needed to include more strains representing different BGC content to explore the hypothesis that strains with higher BGC counts offer more chances of picking up potential biocontrol bacteria. So we have now tested more strains in the dual culture assay, including *Luteibacter* sp. R-72151 that behaved as *Luteibacter* sp. R-73110. We explain the selection in lines 163-177, and the results were added in Table 1.

- What was the criteria used to perform double sequencing strategy. Supplementary table 2 lists genome assemblies that are very heterogeneous. Is there any particular reason why to include in the analysis genomes with more than 100 contigs. BGC analysis of highly fragmented genomes yields unreliable/redundant results. Given the nature of the study, I would like to see the AAI value for each isolate.

We apologise for the lack of explanation. We have now added an explanation of our strategy and summarised why we performed both WGS methods in lines 129-138 (methods), 384-397 (discussion) and Table S1.

In summary, we decided to perform a hybrid sequencing strategy (long + short reads) for some selected strains to obtain full representative, contiguous and complete genomes. Genomes were fragmented based on initial Illumina assemblies, and some produced suspiciously high counts of annotated BGCs with antiSMASH. While we performed the hybrid sequencing for some genomes, we did not include all fragmented genomes (more than 100 contigs) as we did not resequence those whose BGCs seemed to be mostly complete (not truncated or redundant) based on our manual inspection of the antiSMASH results. In any case, as we used relaxed prediction settings in antiSMASH, we expect that in the worst scenario (fragmented genomes), the prediction of BGC counts could be over-represented rather than unrecognised or under-represented.

Regarding the AAI (Average Amino acid Identity) values of the isolates, we did not perform AAI comparison among the isolates but ANI (Average Amino acid Identity), which has a similar application but is more frequently used to compare bacterial genomes (described in Methods, lines 486-488). We do not present detailed info on the values of the comparison because we only used this analysis to figure out if we had duplicated genomes (represented by different isolated strains). None of the included genomes had a similarity higher than 99% (ANI), which denotes there is no duplication of genomes in our dataset, and every strain included in our study is unique. This is now stated in the Results section (line 133). Comparison at the amino acid level would offer less precision than ANI values for the purpose of comparing genome similarities, so we did not include AAI values.

- Line 405. I don't think this conclusion is supported by the data provided by the authors. As mentioned before, there are a few experiments that need to be performed to assure that the genomic analysis provided lead the authors to identify strains with biocontrol activity against the pathogens tested. If authors want to claim that they performed an efficient strain prioritization, they should demonstrate it by testing that strains not selected do not have the relevant bioactivity (using co-culture and crude extract assays).

As mentioned above, we took this criticism on board and critically reviewed our data and how we describe our findings. We performed dual culture assays on additional strains and included these results. It indeed shows the value of genome analysis to prioritise strains for confirmation tests. However, it also showed that a lower proportion of strains with few BGCs can still be of biocontrol value, too, perhaps through novel or specific mechanisms that still require intensive *in vitro* testing in order to be recognised. We have also reworked our Results and Discussion sections to make a more clear argument and explain these conclusions.

Minor comments

- In the abstract section, authors mention that plant -growth promoting activity was tested as part of the study, however there are no results on this regard in the manuscript. Please modify the sentence.

We apologise again for the misunderstanding. Indeed, our study looked only for potential biocontrol features. These mentions were deleted from the abstract.

- Line 187. Please provide a justification for the selection of the fungal strains for the co-culture experiments.

A justification has been added in the final paragraph of the introduction. These fungi are pathogens with a high impact on agriculture worldwide, with broad hosts; therefore, we prioritised them. *Fusarium graminearum*, *Fusarium oxysporum* and *Rhizoctonia solani* are among the most important fungal phytopathogens worldwide; we selected the reference strains. In addition, *Pythium ultimum*, as an oomycete, has a broad host range; we chose one of the three strains available in the BCCM/MUCL Agro-food & Environmental Fungal Collection.

- Line 193. Please provide a sentence describing the overall results obtained in the screening. How many strains showed strong activity? how many strains showed weak activity? and how many strains did not show the expected activity? Also, include this results in the discussion section.

This has been elaborated in the results and conclusion sections, lines 192-197 and 344-351.

- Line 216 and again in line 236. Please remove such sentence unless it is properly described in the strain prioritization process.

We described better our selection process and why genome mining was a useful tool to narrow down bacteria of interest, lines 163-170.

- Line 242. Specify how many strains were selected and why?

Sixty-three strains (Table 1) were initially selected. The selection was based on the biosynthetic potential to produce secondary metabolites encoded by BCGs predicted from annotated genomes and the selection of representative strains of diverse taxonomic groups represented in the collection. We expanded our explanation in lines 163-170.

- Please expand the discussion section by adding arguments regarding the media used to trigger secretion of natural products. What other culture media has been previously used in similar studies? Authors should include in the discussion section that this type of problem can be easily solved by using crude extracts for bioactivity testing.

We have updated the discussion lines 358-371, mentioning how the nutrient composition of the media influences the metabolites produced by the bacteria, and that growing bacteria on a wide diversity of media, collecting their metabolites and exposing the fungi to the extracts can be performed to overcome this limitation. However, we also specify that some antimicrobials are only produced in co-cultivation, and growing bacteria in pure culture can miss this information.

- I recommend authors to perform NRPS analysis (supplementary figure 5) with complementary tools, such as adenylation domain specificity (NRPSpredictor2) or other tools that provide more detailed information.

NRPSpredictor2 webserver has been discontinued from active development and service (<http://nrps.informatik.uni-tuebingen.de/>). However, the algorithm is now used as part of the tools that antiSMASH uses to predict adenylation domain specificity, which has already been included in our results (Figure S5).

- I recommend authors to perform a comparative study for the complete BGCs detected using BigSlice. This will provide an indication of how many different BGCs (based on GCFs) are within the strains selected and how many are potentially novel compounds. In addition, I think Table 2 should be removed as the information provided from AntiSmash pipeline (% similarity) is sometimes not precise. I strongly suggest substituting Table 2 for a more relevant comparative analysis such as BigSlice comparative network using only complete BGCs.

Thank you for the insightful recommendation; it has been implemented in our last version of the manuscript. We performed such a comparative study, particularly for all *Pseudomonas* strains, as we had an acceptable number of species from this genus to perform comparative genomics, and few demonstrated biocontrol effects in our *in vitro* studies.

Here is the process we followed: In the first step, we compared all BGCs predicted by antiSMASH from our *Pseudomonas* strains using BigScape (this package is more appropriate for small-medium size datasets like ours). It is the predecessor of BigSlice, developed by the same group and is actively maintained). We figured out that *Pseudomonas* sp. R-71838 had unique BGCs in our collection, and, since this strain also showed superior biocontrol effects in our assays, we decided to study more deeply these unique BGCs. We wanted to assess if these unique BGCs were novel, considering all genomes sequenced worldwide and deposited in public databases. If not, we wanted to find homologs already described and characterised. To do so, we tried to access the database BigFam (<https://bigfam.bioinformatics.nl/>), with the aim of using it in combination with BigSlice. Unfortunately, we could not get access to BigFam either on the web server or locally at the time of the updates (February-May). Both BigSlice and BigFam are not in active development for a few years since they were published (the database relies on a selection of genomes from NCBI collected until 2020). We noticed that the webserver is back to work recently at the time of this re-submission, but we still believe our alternative approach is valid.

The alternative solution we came up with was using cblaster (<https://cagecat.bioinformatics.nl/>, also developed by the same group led by Dr. Marnix Medema) to retrieve all the similar clusters from the current NCBI database, together with the matches from antiSMASH (module ClusterBlast) and clustered with BigScape to get the closest matches to our unique BGCs from R-71838. This is explained in detail in lines 494-498 (Methods), 278-289 (Results), Figure 4, Figure 5, Table S3 and Table S4.

- Did authors perform the co-culture assay with other isolates that did not pass their prioritization stage. This might be relevant to report in case it was performed. Given what authors describe in the manuscript, I doubt that all the excluded strains do not encode for any antifungal metabolite.

Yes, we did test strains that were very poor in BGCs (had less than 5 BGCs), such as R-73066, R-72562, R-73111, R-72406, and R-71986. We now added R-71875, R-72291, R-72356, R-72369,

R-72421, R-73062, R-74106, R-72435 in the additional tests that were included. With the exception of R-74235 against MUCL 53834, all of them did not show an antifungal effect in the dual culture assay.

June 25, 2023

Prof. Anne Willems
Universiteit Gent
Gent
Belgium

Re: Spectrum00747-23R1 (Uncovering genomic features and biosynthetic gene clusters in endophytic bacteria from roots of the medicinal plant *Alkanna tinctoria* Tausch as a strategy to identify novel biocontrol bacteria)

Dear Prof. Anne Willems:

I am glad to inform you that your manuscript has been accepted for publication in *Microbiology Spectrum*. Thank you for taking into account the recommendations made by both reviewers.

Please make sure to include a "data availability" statement in the final version of your manuscript.

Your manuscript has been accepted, and I am forwarding it to the ASM Journals Department for publication. You will be notified when your proofs are ready to be viewed.

Thank you for submitting your paper to *Spectrum*.

Sincerely,

Frédérique Reverchon
Editor, *Microbiology Spectrum*
